# Seed Treatment with Electromagnetic Field Induces Different Effects on Emergence, Growth and Profiles of Biochemical Compounds in Seven Half-Sib Families of Silver Birch

**DOI:** 10.3390/plants12173048

**Published:** 2023-08-24

**Authors:** Ieva Čėsnienė, Diana Miškelytė, Vitalij Novickij, Vida Mildažienė, Vaida Sirgedaitė-Šėžienė

**Affiliations:** 1Institute of Forestry, Lithuanian Research Centre for Agriculture and Forestry, Liepų 1, LT-53101 Girionys, Lithuania; vaida.seziene@lammc.lt; 2Department of Environmental Sciences, Vytautas Magnus University, Universiteto 10, LT-53361 Kaunas, Lithuania; diana.miskelyte@vdu.lt; 3Institute of High Magnetic Fields, Vilnius Gediminas Technical University, Saulėtekio al. 11, LT-10223 Vilnius, Lithuania; vitalij.novickij@vgtu.lt; 4Department of Immunology, State Research Institute Centre for Innovative Medicien, Santariskiu g. 5, LT-08406 Vilnius, Lithuania; 5Faculty of Natural Sciences, Vytautas Magnus University, Universiteto 10, LT-53361 Kaunas, Lithuania; vida.mildaziene@vdu.lt

**Keywords:** antioxidant enzymes, antioxidant activity, *Betula pendula* Roth, biotic stress, chlorophylls, electromagnetic field, half-sib family, induced resistance, lipid peroxidation, secondary metabolites, seed treatment, sugars

## Abstract

In the context of climate change, strategies aimed at enhancing trees’ resistance to biotic and abiotic stress are particularly relevant. We applied an electromagnetic field (EMF) seed treatment to observe changes in the establishment and content of biochemical compounds in silver birch seedlings induced by a short (1 min) seed exposure to a physical stressor. The impact of EMF treatment was evaluated on seedling emergence and growth of one-year-old and two-year-old seedlings from seven half-sib families of silver birch. The effects on numerous biochemical parameters in seedling leaves, such as total phenolic content (TPC), total flavonoid content (TFC), amounts of photosynthetic pigments, total soluble sugars (TSS), level of lipid peroxidation level, antioxidant activity and activity of antioxidant enzymes, were compared using spectrophotometric methods. The results indicated that, in one-year-old seedlings, two of seven (60th and 73rd) half-sib families exhibited a positive response to seed treatment with EMFs in nearly all analyzed parameters. For example, in the 60th family, seed treatment with EMFs increased the percentage of emergence by 3 times, one-year-old seedling height by 71%, leaf TPC by 47%, antioxidant activity by 2 times and amount of chlorophyll a by 4.6 times. Meanwhile, the other two (86th and 179th) families exhibited a more obvious positive response to EMF in two-year-old seedlings as compared to one-year-old seedling controls. The results revealed that short-term EMF treatment of silver birch seeds can potentially be used to improve seedling emergence and growth and increase the content of secondary metabolites, antioxidant capacity and photosynthetic pigments. Understanding of the impact of EMFs as well as the influence of genetic differences on tree responses can be significant for practical applications in forestry. Genetic selection of plant genotypes that exhibit positive response trends can open the way to improve the quality of forest stands.

## 1. Introduction

Forests cover 30% (about 3.8 billion ha) of the earth’s surface, are rich in biodiversity and provide benefits to humans such as clean air and water, timber, fiber and fuel [1,2]. Therefore, environmental devastation, loss of biodiversity and climate change are the main ecological threats to European forests. In order to overcome these challenges, the European Green Deal was presented in December 2019 [3]. The main goal of this action plan is to turn the challenges listed into new opportunities for the environment and climate-friendly, sustainable development. Forestry development strategic documents and legal enactments emphasize the aspiration to restore forests on a genetic–ecological basis (selectively valuable high-quality forest-propagating material). An integral part of this process is the implementation of selection in the forest, increasing the resistance of trees against abiotic and biotic stress factors. Using planting material characterized by good growth, wood quality and resistance to diseases and pests can be the main solution to increase the productivity and sustainability of stands [4].

About 70% of forests are vulnerable to diseases caused by pathogenic fungi [5,6,7]. In general, tree pathogens are responsible for economic losses that usually amount around USD 2.1 billion worth of forest products are lost annually in the United States due to invasive forest pathogens [5]. In the last few decades, more attention has been paid to looking for safer non-chemical-based and effective technologies not only for agriculture but also for sustainable forestry. These modern solutions could become an alternative to the success of the European Green Deal strategies. 

The development of innovative technologies for improving germination, seedling growth and resistance against abiotic and biotic stress factors has recently reached high importance. The use of physical techniques such as non-thermal plasma [8], electromagnetic fields (EMFs) [9] and ultrasound [10,11] have received much attention, since these methods are considered safer and more environmentally friendly. It has been reported [12] that both magnetic and electric fields can improve the seed quality of numerous crops. The positive impacts of such treatments include better seed germination and seedling growth as well as higher yields. Seedlings obtained from treated seeds are more resistant to unfavorable environmental conditions [13]. For various agricultural crops, short-term seed treatment with electrical discharge [14,15,16] or electrostatic or pulsed electromagnetic fields [17,18,19] also resulted in improved seed quality (e.g., germination increased up to 20% in most cases). Previous studies showed that seed treatment with physical stressors can be an effective for increasing plant resistance to various pathogens, as EMFs have been found to decrease microbial seed decontamination [20,21,22,23]. In addition, it was noted that seed treatment with EMFs led to increased content of secondary metabolites (SMs) including phenolic compounds in *E. purpurea* seedlings [24]. The germination rate of Norway spruce seeds was efficiently stimulated by EMFs, as was the early growth of seedlings (increased by 50–60%, compared to control) [25].

Pre-sowing seed treatment with physical stressors, such as EMFs, could be used to induce the systemic resistance in plants. Induced systemic resistance (ISR) is a protective strategy of plants, through which plants can protect themselves from infections caused by pathogens, pests, etc. [26]. ISR can modify and enhance positive plant responses to stress by leveraging natural defenses, such as increased polyphenol levels, heightened antioxidant enzyme activity and improved photosynthetic pigments [26,27,28,29,30,31]. Some studies showed that ISR has been associated with decreased severity of insect attacks on pine trees [32], as well as diminished disease severity in cucumber plants infected with the *Pythium aphanidermatum* pathogen [26,33]. Several significant mechanisms could be categorized as instances of induced systemic resistance. One of them involves the stimulation of the synthesis of SMs, which are the plant’s natural defenses [26,30,34]. Significant correlations between SMs in Norway spruce needles and plant resistance to pathogens were reported previously [35]. Elevated SMs, as well as enhanced activity of antioxidant enzymes such as catalase (CAT), peroxidase (POX), ascorbate peroxidase (APX) and glutathione reductase (GR) [36,37,38] and synthesis of protective structural barriers [26,27,28,29,30] are outcomes of the activated natural plant defense system. It is known that such defense mechanisms are related to the ability of plants to protect themselves from infection/infestation (pathogens, herbivores, etc.) through indirect inhibition of the invader or to respond to other types of stress [27,28,29,30,31]. These mechanisms are called induced systemic resistance (ISR). It was noted that exposure to magnetic fields (MF) can lead to reduced plant oxidative damage due to the enhanced activities of antioxidant enzymes such as peroxidase, polyphenol oxidase (PPO), superoxide dismutase (SOD) and catalase (CAT) in plant cells. Specifically, MF affected the antioxidant activity and increased the activity of the free radicals in plants [39,40,41]. The non-enzymatic components of the antioxidant defense system are small ROS scavengers, such as ascorbate, glutathione and certain photosynthetic pigments, e.g., carotenoids [36,37,38,42,43]. Plants indirectly respond to stress by changing amounts of other photosynthetic pigments, i.e., chlorophylls [44,45,46]. However, information about the formation of induced systemic resistance (ISR) of silver birch (*Betula pendula* Roth) seedlings grown from seeds treated with EMFs as a physical stressor is not available.

The silver birch is a fast-growing, light-demanding, widely distributed and economically important deciduous tree in Northern Europe [47,48,49]. In Lithuania, silver birch occupies 17% of the total forest area and is important for the ecological sustainability of the forests, as well as for the social and economic sectors [50]. The leaves of silver birch are used to treat rheumatic disease, nephrolithiasis, prostatitis and atherosclerosis, induces perspiration and possesses diuretic and anti-inflammatory effects [51]. The metabolic profile of *B. pendula* is known to vary with genotype and latitude of origin across Northern Europe [52]. Due to climate change, silver birch is becoming more susceptible to pathogens. These trees are attacked by the primary pathogens *Phytophthora cactorum* (Lebert and Cohn) and *Marssonina betulae* (Lib.) [53,54]. Over the past few decades, there has been a noticeable increase in attention towards *Phytophthora* species acting as plant pathogens in Europe and globally [55,56,57,58,59]. Additionally, the aggressive pathogen *Marssonina betulae* induces the development of sunken stem cankers and progressive crown dieback in the silver birch [60,61]. Various pathogens, along with other factors such as poor silvicultural management, unsuitable birch provenance, incorrect site selection and climate change, contribute to birch dieback, making trees more susceptible to disease [60]. Therefore, it is crucial to seek solutions that will help maintain the health of trees. Earlier studies also showed that different tree genotypes synthesize different amounts of SMs and mobilize them against pathogens; therefore, genetic selection based on this trait is one of the most important factors in these kinds of studies [4,62,63].

In the previous study, the effects of cold plasma (CP) and EF treatments on Norway spruce germination and growth were investigated using a natural seed mixture [25,64]. It was demonstrated in the later studies on seven half-sib families that effects of seed treatments on growth and biochemical composition of Norway spruce seedlings are genotype-dependent [34,65]. The selection of more vigorous and resistant silver birch genotypes can be recommended to forestry companies for effective reduction of pathogen losses. In this study, our aim was to compare the response of genetically diverse silver birch families to pre-sowing seed treatment with EMFs. We measured the effects on the amounts of antioxidant enzymes, antioxidant activity, photosynthetic pigments, secondary metabolites, lipid peroxidation and total soluble sugars in the leaves during the first two vegetation seasons. There is a lack of research to offer more distinct comprehension of how short-term physical stressors can affect woody plants over a long-term period. The majority of publications are focused on the impact of stressors on germination and the height of early-stage plants. A few reports marked differences in percentage of germination and seedling growth between early and long term in woody plants [25,66]. Based on these studies, the effect of seed treatment persists for a duration longer than merely the early stages of plant development. However, there is a gap in the research about the synthesis of biological compounds in woody plants over a long-term period. The objective of our study was to gather information about the potential positive impact on plant resistance to pathogens. Similar systematic studies of pathogen resistance in tree planting material through pre-sowing seed treatment by physical stressors have not been conducted so far.

## 2. Results

### 2.1. EMF Effects on Emergence and Growth of Silver Birch Seedlings

The emergence percentage of silver birch seedlings in the control groups was low in all half-sib families, ranging from 2.31 (60th family) to 16.67 (73rd family) percent. The emergence of silver birch was dependent on half-sib family and seed treatment with EMFs increased the emergence in five out of seven half-sib families (Table 1).

Seed treatment with EMFs did not affect the emergence in the 73rd family, which was characterized by the highest emergence percentage both in the control and treated groups (Table 1). The highest positive EMF effect (more than 3 times) on seedling emergence was observed in the 60th family, with the lowest emergence percentage in the control. Emergence was also stimulated in the 171st, 112th and 179th families (89%, 85% and 31%, respectively), while the positive effect (14%) was less pronounced in 86th family. However, seed treatment with EMFs resulted in 35% lower emergence in the 125th family. Thus, effects of EMF treatments on the emergence of silver birch seedlings were genotype-dependent. 

The growth of seedlings in the first and second vegetation season was estimated by measuring the height of the above-ground part of the seedling (Figure 1). In the first year, the control seedlings of the 171st family were highest, compared to other families. Meanwhile, the control seedlings in the 86th and 179th families were twice as short compared to those from the 171st family. Seed treatment with EMFs significantly increased the height in four half-sib families: 60th, 86th, 112th and 125th (by 71%, 43%, 22% and 39%, respectively). However, there was no difference between the height of one-year-old seedlings in control and EMF-treated groups for the 73rd, 171st and 179th families.

All differences between the one-year-old and two-year-old seedling height in the same experimental groups were statistically significant (Figure 1). The two-year-old control seedlings were highest in the 73rd and 112th families, and lowest in the 86th family. Among the EMF-treated groups, the highest silver birch seedlings were also in the 73rd and 112th families. Furthermore, EMF exposure induced a statistically significant height increase in three half-sib families (by 22%, 41% and 23% in 60th, 86th and 171st family, respectively). The height of seedlings in the 73rd and 179th family was not affected by the EMF, like for the one-year-old seedlings of these families, but the height of two-year-old seedlings in 171st family was 23% larger compared to the control. In the 86th family, one-year-old seedlings in the EMF-treated group were 43%—and two-year-old seedlings, 41%—larger compared to the control. Thus, the dynamics of EMF effects on seedling height can be variable, such that in some families the effects can be observed later than in others.

In summary, a short duration (1 min) of treatment of silver birch seeds can stimulate germination and early seedling growth, but the observed effects strongly depend on the half-sib families. Out of seven half-sib families, considerable stimulation of germination was determined in four families (60th, 112th, 171st, 179th) and at least two families (60th and 86th) exhibited improved height parameters in both one-year-old and two-year-old seedlings. In contrast, the height of seedlings in two families (73rd and 179th) was not affected by EMF treatment.

### 2.2. EMF Effects on Activities of Antioxidant Enzymes

The results of the performed measurements of antioxidant enzymes (SOD, POX, APX, CAT, GR) in silver birch leaves are presented in Figure 2, Figure 3, Figure 4, Figure 5 and Figure 6.

The results showed high variation of SOD activity in leaves between half-sib families of silver birch (Figure 2). In one-year-old seedlings, the 60th family had the highest activity of SOD in the control compared to other families. The difference in SOD activity between the 60th family and 86th or 179th families with the lowest SOD activity was 4.2 and 3.6 times, respectively. Similar variations among families were observed in EMF-treated groups. Among seedlings growing from EMF-treated seeds, the 171st family had the highest SOD activity compared with other families. The treatment with EMFs significantly increased activity of SOD in two half-sib families: the 125th and 179th (by 5 and 78%, respectively). However, in four families (73rd, 86th, 112th, 171st), EMFs did not change SOD activity, and even decreased it by 61% in the leaves of the 60th family.

SOD activity was higher in two-year-old seedlings when compared to one-year-old silver birch seedlings. In two-year-old seedlings, SOD activity was highest in the 171st family, both in the control and EMF-treated groups, compared to other families. Meanwhile, the 73rd and 179th families had the lowest activity in control. Seed treatment with EMFs had a positive effect on SOD activity in four half-sib families: the 73rd, 125th, 179th and 171st (activity increased by 92%, 39%, 19% and 16%, respectively, compared to control). In both sample collection years, the activity of SOD was reduced after seed treatment with EMFs for the 60th family only. Meanwhile, in both years, the activity of SOD increased after seed treatment for two families: the 125th and 179th.

The highest POX activity in the control groups of one-year-old seedlings was observed in the 171st family compared to other families (Figure 3). The lowest activity was determined as the 73rd family (the difference in POX activity between the 171st and 73rd families was 1.8 times). The 171st family had the highest activity after seed treatment with EMFs. Seed treatment with EMFs increased the POX activity in two half-sib families: the 73rd and 179th (by 117% and 17%, respectively). An EMF-induced decrease in POX activity was observed in the 60th and 112th families (12% and 41%, respectively), while EMF treatment had no effect in the 86th, 125th and 171st families.

The highest POX activity in the control group of two-year-old seedlings was detected in the 112th family, and it exceeded the lowest activity found in the 179th family by 2.4 times. An EMF-induced increase in POX activity was established in three half-sib families: the 73rd, 86th and 179th (the activity increased by 22%, 57% and 84%, respectively, compared to control). A significant EMF-induced decrease in POX activity was observed in the 112th family, whereas, for the 60th, 125th and 179th families, the EMF effect was neutral.

Finally, we conclude that seed treatment with EMFs led to an increase in POX activity in two families, the 73rd and 179th, in both one-year-old and two-year-old silver birch seedlings.

Similar activity of APX, ranging from 42 to 43 µmol/mg, was observed in the leaves of one-year-old control seedlings in the 112th, 125th and 179th families (Figure 4). In contrast to these families, APX activity detected in the 73rd family was almost 3 times lower. After seed exposure to EMFs, the highest APX activity was observed in the 125th family. EMF treatment increased the APX activity in two half-sib families: the 73rd and 86th (increased by 45% and 27%, respectively, compared to the control). EMF treatment slightly decreased APX activity in the 60th and 112th families, and did not induce changes in the 125th, 171st and 179th families.

The activity of APX was higher in two-year-old seedlings compared to one-year-old seedlings (except the 86th family in the control). In two-year-old seedlings, the highest APX activity was observed in the control group of the 125th family and the lowest activity was determined as the 73rd family (with an 8.9 times difference). After exposure to EMFs, the highest APX activity was found in the 86th family; it was increased by EMF treatment by 5.5 times. Short-term seed treatment with EMFs also increased activity of APX in three half-sib families: the 60th, 86th and 179th (by 25%, 360% and 29%, respectively, compared to the control). In contrast, a strong decrease (by 69%) in APX activity was induced by EMFs in the 125th family. EMFs had no effect on APX activity in the leaves of two-year-old seedlings from the 73rd, 112th and 171st families. The obtained results indicate that seed treatment with EMFs has the potential to increase the activity of APX in the silver birch leaves; however, the effect is dependent on the half-sib family.

The highest CAT activity in the control group of one-year-old seedlings was observed in the 60th family while the lowest activity was observed in the 73rd family (difference of 1.5 times) (Figure 5). The highest CAT activity after seed treatment with EMFs was found in the 86th family. Seed treatment with EMFs increased CAT activity in three families: the 73rd, 171st and 179th (by 50%, 27% and 8%, respectively, compared to the control). In contrast, CAT activity in the seedlings of the 112th family was reduced (by 37%) by EMF treatment and remained the same as the control in the 60th, 86th and 125th families.

CAT activity was generally higher in two-year-old seedlings compared to one-year-old seedlings in all half-sib families (except the 86th and 179th families in the control group). Among the control groups of two-year-old seedlings, the highest CAT activity was detected in the 112th and 125th families, while the lowest activity was in the 179th family (a difference of more than two times). Moreover, the highest enhancement in CAT activity (by 85%) after seed treatment with EMFs was determined in the 86th family. In addition, seed treatment with EMFs increased CAT activity in the 179th family (by 34%). However, EMF treatment decreased CAT activity in the 112th family but did not induce changes in the 60th, 73rd and 71st families. Only one family (179th) had increased CAT activity in both the one-year-old and two-year-old seedling samples.

The highest GR activity among the control groups of one-year-old seedlings was determined in the 60th and 86th families, while the 73rd family had the lowest GR activity (largest difference in GR activity was 1.6 times) (Figure 6). On the other hand, the 171st family had the highest GR activity after seed treatment with EMFs compared with other families. The treatment with EMFs increased GR activity (40%) in the 73rd family only. The EMF-induced decrease in GR activity was found in the 60th and 112th families, while activity did not change in the rest of the families. GR activity was higher in two-year-old seedlings compared to one-year-old seedlings in all half-sib families (except the 86th family in the control group).

In the control samples of two-year-old seedlings, the GR activity was highest in the 125th family compared to other families. Meanwhile, the 73rd family had the lowest activity in the control group. The strongest stimulation of GR activity (94%) by seed treatment with EMFs was observed in the 86th family, but a significant increase was established in two other half-sib families: the 60th and 179th (activity increased by 11 and 41%, respectively, compared to the control). The effect of EMF treatment in the remaining four half-sib families was neutral.

In conclusion, the obtained results revealed that short-term EMF treatment has the potential to alter the activity of antioxidant enzymes; however, these induced changes are strongly dependent on half-sib family and the age of the seedlings. For example, EMF treatment decreased activities of four out of five antioxidant enzymes in one-year-old seedlings of the 60th and 112th families, while opposite effects (increased activities of four or three out of five enzymes) were observed in the 73rd and 179th families, and only one out of five enzymes were stimulated in the 86th (APX), 125th (SOD) and 171st (CAT), whereas activities of other enzymes remained unaffected. In general, in the leaves of 2-year-old seedlings, the measured activities of antioxidant enzymes (except POX) were higher compared to the one-year-old seedlings. In addition, EMF effects on their activities were different both qualitatively and quantitatively in some of the families. For example, similar to the first year, POX and APX activity was decreased by EMF in the 112th family. In contrast to the first year, some enzymes (GR and APX) were stimulated in the 60th family. Rapid growth of seedlings in the EMF-treated group in the 86th family was followed by a strong increase in POX, APX, CAT and GR, which was observed in the second year of vegetation only.

### 2.3. Antioxidant Activity

The antioxidant activity in the leaves of seven half-sib families of silver birch was measured by two radical scavenging tests: DPPH (2,2-diphenyl-1-picryl-hydrazyl-hydrate) and ABTS (2,2′-azino-bis(3-ethylbenzothiazoline-6-sulfonic acid). The results obtained in samples of control seedlings showed that antioxidant activity was dependent on the half-sib family (Table 2). DPPH radical scavenging activity variated between 225 µmol/g (73rd family) and 1025 µmol/g (171st family), while differences were much less pronounced for the ABTS test, although the same families manifested minimal (337.8 µmol/g in 73rd family) and maximal values (686.9 µmol/g in 171st family) of antioxidant activity in the leaves of one-year-old seedlings. Both tests indicated an increase in antioxidant activity in the second year of vegetation of control seedlings. The magnitude of that increase was also dependent on the half-sib family. DPPH scavenging activity in the two-year-old seedlings was 1.28 times (86th family) to 2.44 times (73rd family) larger compared to the one-year-old seedlings of the same family. The only exception was noticed in 171st family, where DPPH scavenging was the highest in the first year but decreased by 67% in the second year.

However, the results of the ABTS test for this family followed the same trend as in all other families—a strong increase in antioxidant activity (from 1.8 times in the 179th family to 3.4 times in the 73rd family) was found in the two-year-old control seedlings compared to the one-year-old ones. Furthermore, the results obtained showed that seed irradiation with EMFs can increase antioxidant activity in silver birch leaves (Table 2). In the one-year-old seedlings, DPPH radical scavenging activity was almost doubled by EMFs in the 60th family; a moderate positive EMF effect (14%) was observed in the 125th family. EMFs had no effect on DPPH scavenging in the 73rd and 86th families and strongly (3.2 times) reduced this activity in the 171st family. ABTS scavenging activity was increased (1.6 times) by EMF treatment in the 60th and 73rd families, reduced (by 30%) in the 86th family and not affected in the 112th, 125th, 171st and 178th families. In contrast to the one-year-old seedlings, seed treatment with EMFs had a much smaller effect on antioxidant activity in two-year-old seedlings (characterized by much higher antioxidant activity in control groups). In all half-sib families, EMFs had no effect on DPPH scavenging activity, except for slightly inhibited (8%) activity in the 112th family. ABTS scavenging was not affected by EMFs in the 60th and 125th families, stimulated (by 13%) in the 86th family, and inhibited to various degrees (from 5% to 22%) in the remaining four families.

Thus, the obtained results indicate that the EMF effect on antioxidant activity in leaves of silver birch is strongly dependent both on half-sib family and age of seedlings.

### 2.4. EMF Effect on the Amount of Photosynthetic Pigments

The results of spectrophotometric detection of chlorophylls and carotenoids in leaf extracts of silver birch showed that, among the one-year-old control groups, the highest content of chl a and chl b was found in the 125th family, and the lowest content was observed in the 60th family (the difference between these two families in the amounts of both chlorophylls was 2.3 times) (Table 3). Meanwhile, the amount of carotenoids in the control seedlings of seven half-sib families was rather similar (varied between 14.01 and 17.36 μg/g). Seed exposure to EMFs induced a significant increase in chl a, chl b and caro were observed in three half-sib families, the 60th, 73rd and 179th families, in one-year-old seedlings. However, in the 112th family, an increase in chlorophylls and decrease in carotenoids was noted. Additionally, in two families (86th and 125th families), the content of chlorophylls was slightly decreased due to the effect of EMF exposure (with no changes in carotenoids content).

The amounts of pigments in the two-year-old control seedlings increased compared to one-year old seedlings of the same half-sib families (Table 3), except for the decrease in the amount of chl a in the 86th, 125th and 171st families and unchanged amount of chl b in the 171st family. The amount of carotenoids increased substantially in the second vegetation season (from 2.3 times in the 86th family to 4.6 times in the 60th family), and differences in caro amount among families became more evident. An EMF-induced increase in the amount of chlorophylls was observed in five families (60th, 73rd, 86th, 171st and 179th), with the largest effects in the 60th family (chl a content increased by 2.5 times, chl b by 92%). Moreover, the 171st family exhibited an increase in all photosynthetic pigments. However, seed exposure to EMF decreased the content of carotenoids in the 60th and 125th families (by 32% and 18%, respectively).

Thus, the obtained results indicate that the accumulation of photosynthetic pigments is dependent on both the half-sib family and the age of the seedlings. Specifically, we observed an increase in chlorophylls in three families (60th, 73rd and 179th) in both one-year-old and two-year-old seedlings compared to the control.

### 2.5. Secondary Metabolites

#### 2.5.1. Total Polyphenol Content (TPC)

Substantial variation in the TPC of leaves of the control one-year-old silver birch seedlings was found among half-sib families (Figure 7). The difference between the largest TPC (86th family) and the smallest (73rd family) was 1.86 times. Seed treatment with EMFs for 1 min induced different TPC changes in the leaves of half-sib families. A significant positive EMF effect on TPC was observed in two families: the 60th and 73rd (TPC increased by 47% and 60%, respectively). The TPC increased by 1.19 mg/g and 1.57 mg/g, respectively, compared to control groups. In contrast, EMFs induced a TPC decrease in the 112th and 179th families (by 8% and 15%, respectively). In three families (86th, 125th and 171st), EMF treatment had no effect on TPC.

The obtained results revealed that two-year-old seedlings accumulate more phenolic compounds compared to one-year-old silver birch seedlings. The highest TPC in the control groups of two-year-old seedlings was in the 125th family and it exceeded by two times the lowest TPC (similar to the one-year-old seedlings in the 73rd family). Seed treatment with EMFs resulted in a TPC increase only in the 179th family (by 8%), while EMF did not affect TPC in the 60th, 73rd and 86th families and decreased TPC in the 112th, 125th and 171st families (by 26%, 31% and 12%, respectively).

In summary, the results demonstrated an increased TPC in two-year-old seedlings compared to one-year-old seedlings. Changes in TPC induced by seed treatment with EMFs were dependent on half-sib family, and strong positive EMF effects were found in in two families (60th and 73rd) among the one-year-old silver birch seedlings, but seedlings from only one family (179th) exhibited a slight increase in the levels of these compounds in the second year of vegetation.

#### 2.5.2. Total Flavonoid Content (TFC)

Among the one-year-old control seedlings, the highest TFC was determined in the 86th and 171st families and it was approximately 91% higher compared to the lowest value determined in the 60th family (Figure 4). Furthermore, the irradiation of seeds with EMFs resulted in a significant increase in TFC in three families: the 60th, 73rd and 86th families (by 60%, 67% and 11%, respectively).

The results showed that accumulation of TFC was higher in the control groups of all families in two-year-old seedlings compared to one-year-old seedlings (except the 86th and 171st families) (Figure 8). The highest TFC among the control groups was observed in the 125th family and it was 2.8 times larger compared to the lowest TFC in the 171st family. After seed treatment with EMFs, the highest TFC was also observed in the 125th family, similar to that in the control. A positive effect of seed treatment with EMFs on TFC was observed in the 73rd, 112th and 179th families (TFC increased by 29%, 25% and 38%, respectively). EMFs had no effect on TFC in the remaining four families.

Thus, the effect of EMF was found to be dependent on the half-sib family. We observed an increase in TFC in the 73rd family both in one-year-old and two-year-old seedlings of silver birch.

### 2.6. Lipid Peroxidation

Lipid peroxidation was estimated by the amount of malondialdehyde (MDA) in silver birch leaves (Figure 9).

MDA levels in the one-year-old control seedlings varied between 68.6 (in 179th family) and 104.2 nmol/g (in the 125th family). The lowest lipid peroxidation levels (62 nmol/g) after seed treatment with EMFs were established in the 112th and 179th families. (61.91 nmol/g and 62.63 nmol/mg, respectively). EMFs decreased lipid peroxidation in the 86th, 112th and 125th families (by 9%, 21% and 37%, respectively). However, in two families (73rd and 171st), the MDA level was slightly higher in EMF-treated groups compared to the controls. EMFs did not have an effect on lipid peroxidation in the 60th and 179th families.

Compared to the one-year-old control seedlings, the MDA amount in leaves of the two-year-old seedlings was higher in three families (112th, 125th and 179th) but lower in four other families (60th, 73rd, 86th and 171st). The highest level of lipid peroxidation was observed in the 125th half-sib family and it exceeded the lowest level (found in the 86th family) by 33%. Furthermore, seed treatment with EMFs resulted in lower levels of lipid peroxidation in five half-sib families: the 60th, 73rd, 112th, 125th and 179th families (by 12%, 9%, 18%, 14% and 13%, respectively), compared to the control groups. In the remaining two families, MDA amount was not changed by EMF treatment.

Thus, the obtained data indicate MDA levels in silver birch leaves are dependent on seedling age and half-sib family, and that seed treatment with EMFs may reduce levels of lipid peroxidation in silver birch leaves across five out of seven half-sib families.

### 2.7. Total Soluble Sugars (TSS)

Our results showed that the content of TSS in seedlings of different half-sib families of silver birch exhibits significant differences (Figure 10). The highest amount of TSS among seedlings of the control groups was observed in the 179th family, and it exceeded the lowest amount detected in the 73rd family by 70%.

Meanwhile, after seed treatment with EMFs, the 171st family exhibited the highest amount of TSS compared to other families. Seed treatment with EMFs resulted in a significant increase in TSS in the 73rd family (by 41%) only, while decreased TSS compared to the control was detected in the 112th, 125th and 179th families (by 18%, 21% and 44%, respectively). The obtained results indicated that the accumulation of TSS was higher in two-year-old silver birch seedlings compared to one-year-old seedlings (except the 86th family in the control group) (Figure 10). Among the two-year-old control seedlings, the highest TSS level was observed in the 125th half-sib family, and the lowest in the 86th family (difference of 2.5 times). A strong positive effect of seed processing with EMFs was observed in three half-sib families: the 86th, 171st and 179th (by 70%, 55% and 34%, respectively, compared to the control groups). An EMF-induced decrease in TSS of 39% was determined in the 125th family, and in the three remaining half-sib families, EMF had no effect on TSS.

Thus, our data revealed that TSS amounts and EMF-induced changes in silver birch leaves are dependent on the half-sib family as well as on seedling age. Notably, one family, specifically, the 179th family, exhibited an increase in TSS after seed treatment with EMF in both the one-year-old and two-year-old leaves samples.

The summary of the obtained experimental results is provided in the Table 4. Based on the results, we conclude that EMF-induced changes in numerous parameters of silver birch are dependent on the half-sib family. We found that the 60th, 73rd, 86in two-years-old seedlings) and 179th families exhibited a positive response to EMF treatment with changes in the majority of the estimated parameters. However, the EMF had a negative impact on most parameters in the 112th family, and neutral or negative effects dominated in the 125th and 171st families.

## 3. Discussion

There is a knowledge gap in our understanding of the biochemical and physiological processes in silver birch seedlings grown from seed after treatment with EMFs. This manuscript presents novel findings on how seed treatment with EMFs influences seedling emergence and early growth across different half-sib families. The study also examines changes in metabolic and defense enzyme activity, secondary metabolism and photosynthesis, which serve as key indicators of plant adaptability to biotic and abiotic stress. Numerous studies (reviewed recently in [67]) have noted that seed treatment with EMFs increases the seed germination rate in *T. pratense* L. [68], *Triticum aestivum* [69] and woody plants such as *Rhododendron smirnowii, Morus nigra* or *Picea abies* [25,66]. The germination rate in some cases increased by almost 50%, depending on the duration of the treatment with EMFs. Pulsed electromagnetic fields have been found to promote germination and improve early growth characteristics of other plants [70,71,72]. Our findings partially confirmed these applications, as the results showed that EMF positively affected both emergence and growth of seedlings depending on silver birch half-sib families. The 60th half-sib family exhibited a 3.4 times higher seedling emergence after seed exposure to EMFs; this increased for four other families, as well (86th, 112th, 171st and 179th). The same half-sib families of silver birch also showed better one-year-old and two-year-old seedling growth indicators. In addition, the selection of optimal combinations of magnetic field frequency, density and exposure time has been observed to have an impact not only on pigment content but also on enzyme activities [73]. It was reported that treating seeds with a magnetic field (MF) increases the activity of enzymatic antioxidants, such as SOD, CAT and GR in *Brassica juncea* seedlings [74]. Moreover, researchers have noted that seed treatment with EMFs can enhance the dry mass and root and shoot length of *Physalis alkekengi* seedlings. Additionally, positive effects on SOD, POX and CAT activity have been observed [75]. The increase in SOD activity may stem from soluble carbohydrates, including changes in gene expression. In addition, soluble carbohydrates could increase PAL activity, consequently fostering an augmented synthesis of phenolic compounds through the phenylpropanoid pathway [76]. The increased levels of total soluble carbohydrate content, SOD and PAL activity and TPC demonstrate the interrelation among several components involved in inducing the plant defense system [77,78]. Notably, research has demonstrated that a higher content of soluble sugars has the potential to increase plant tolerance against abiotic stresses [79]. Moreover, the amount of TSS is directly dependent on photosynthesis. Sugar production is the most vital process occurring in a plant through the photosynthetic pathway and it is regulated by both internal and environmental factors that impact plant growth and development [80,81]. Photosynthetic pigments, such ascarotenoids and chlorophylls, play distinct roles in plant physiology [82]. Chlorophylls are related with plant viability, while carotenoids are more closely linked to plant stress responses, antioxidative activity and phytohormone production [83,84,85]. The results obtained from this study demonstrated that short-term seed treatment with EMFs can modulate the antioxidant system in silver birch seedlings. The majority of enzymatic antioxidants, such as SOD, POX, APX, CAT and GR, increased in the seedlings of three half-sib families (73rd, 179th and 86th). Additionally, moderate or strong correlations were determined between chlorophyll a and b concentrations and seedling emergence in the 86th, 171st and 179th half-sib families (ranging from r = 0.387 to r = 0.995). Furthermore, a positive correlation between chlorophyll a and b contents and TSS was established in the 86th half-sib family (r = 0.816 and r = 0.539, respectively). The content of carotenoids was related to the changes in TPC. The seedlings of half-sib families, especially the two-year-old seedlings, with higher levels of carotenoids exhibited lower TPC. Genetic families differ in the ability to exhibit peculiar metabolic response to abiotic stress; some of them prioritize the production of phenolics as their primary defense mechanism, while others prefer production of carotenoids. Both TPC and carotenoids have different antioxidant properties. Srivastava noted [86] that carotenoids have the ability to quench singlet oxygen and scavenge toxic free radicals, thereby preventing or reducing damage to living cells. Meanwhile, the phenolic compounds exhibit antioxidant activity, such as the chelation and free radical scavenging, exerting a specific impact by eliminating hydroxyl and peroxyl radicals, superoxide anions and peroxynitrites [87,88]. The studies conducted by other authors have demonstrated that pre-treatment with EMFs enhances the content of TPC and improves membrane stability in *P. alkekengi* seedlings [75]. Guderjan et al. [89] and Chaharsoughi et al. [90] have indicated that pre-treatment with pulsed electric fields and EMFs increased TPC in the *Brassica napus* and *Portulaca oleracea* plants. It has been reported that phenolic compounds exhibit antioxidant properties in vivo by modulating lipid peroxidation [91,92]. It was observed that seedlings grown from seeds treated with physical stressors, such as non-thermal plasma, contained reduced levels of MDA, which is associated with improved germination and growth. Such findings suggest that short-term seed treatment with stressors could potentially alleviate membrane damage by enhancing the antioxidant system of the plants, and that leads to achieving positive effects on seedling development [93,94,95,96,97].

The antioxidant activity of flavonoids depends on the arrangement of functional groups within their structure. It has been reported that the flavones and catechins seem to be the most powerful among flavonoids in protecting the plant cells against ROS [98]. Therefore, these antioxidant compounds play a pivotal role in enhancing plant stress tolerance [99]. It has been reported that seed treatment with EMFs leads to an increase in TFC in *Portulaca oleracea* [90]. In addition, the results of our previous studies indicated that seed processing with cold plasma induced changes in TFC as well as TPC, photosynthetic pigments and antioxidant activity in the needles of different Norway spruce half-sib families [34,35]. The short-term seed exposure to EMF elicited notable enhancements in TFC/TPC in one-year-old seedlings of two silver birch half-sib families, the 60th and 73rd. Additionally, a significant elevation in antioxidant activity was observed in these families. However, there was no correlation established between TFC/TPC and antioxidant activity in two-year-old seedlings.

Notably, all the induced changes were obviously dependent on the specific half-sib family. The seven studied silver birch families can be categorized into several groups based on their response to seed treatment with EMFs. The first group, comprising four families (60th, 73rd, 86th and 179th), is characterized by the prevailing positive changes in most of the estimated parameters. Among these four families, a strong positive response was characteristic for seedlings of the 60th and 73rd families in the first year, which became moderately positive in the second year of vegetation; meanwhile, the positive responses of seedlings in the 86th and 179th families were more obvious in the second year compared to the first year of vegetation. In two other half-sib families (112th and 125th), most of the changes induced by EMFs were negative, while the response of seedlings in the 171st family was neutral (one-year-old) or moderately positive (two-year-old seedlings). The results obtained from this study reveal that short-term (1 min) treatment of silver birch seeds with EMFs elicits a complex response, including changes in seedling performance (emergence and growth) as well as in numerous biochemical parameters relevant to seedling physiological and biochemical processes, along with defensive capacity and stress tolerance. The forthcoming investigations associated with the outcomes of this article involve inoculation of these trees with specific pathogens, seeking to assess differences in both untreated and treated silver birch tree samples. These studies will afford us the opportunity to discern a response to direct biotic stress and to appraise alterations in bioactive compounds over long-term studies.

## 4. Materials and Methods

### 4.1. Planting Material

Seeds of seven different half-sib families of silver birch (*Betula pendula* Roth.) were collected from a second-generation birch seed orchard in the Dubrava regional division (Figure 11).

### 4.2. Seed Treatment with EMFs and Cultivation of Seedlings

Seeds of each silver birch half-sib family were exposed to radiofrequency electromagnetic field (EMF) treatment using a pulsed magnetic field generator (peak parameters 100 kHz sine-wave, 0–10 mT) designed at Vilnius Gediminas Technical University (Figure 12).

Seeds were treated for 1 min using a 100 kHz, 400 ± 50 μT oscillating magnetic field (other conditions include room temperature, atmospheric pressure). For each experimental group, 216 seeds were treated at one time, keeping them 3 cm above the center of the induction coil. Non-treated seeds were used as a control. The total number of seeds used for this study was 3024 (7 genotypes × 2 groups × 216 seeds). After a one-time treatment, the seeds were left at room temperature for 4 days to provide time for changes in the content of seed phytohormones [100]. Earlier experiments conducted on radish seeds demonstrated that seed treatments induce rapid changes in the amounts of phytohormones (abscisic acid and gibberellins) which play a role in germination. The most favorable GA/ABA ratio was achieved four days after seed treatment.

Four days after treatment, control and EMF-irradiated seeds were sown in blocked randomized cassettes filled with a peat substrate (pH of 5.5–6.5) and covered lightly with perlite (Figure 13A). Eighteen-cell cassettes with cell dimensions of 6 × 6 × 13 cm were used. For the first two months, the emerged seedlings were grown under controlled conditions in a greenhouse where temperature was maintained between 25–32 °C during the summer days and did not drop below 10 °C at night. Two months after sowing, seedlings were transferred outdoors to an open area with abundant natural light. After one year, the seedlings were planted in larger pots (18 × 17 × 2 cm) filled with 1000 mL peat substrate (SuliFlor SF2) (Figure 13C).

### 4.3. Evaluation of Seedling Emergence and Morphological Parameters

Emergence. The emerged seedlings of silver birch were counted beginning 7 days after sowing, and counting was continued for 15 days (until the number of the emerged seedlings did not change). Then, the percentage of emerged seedlings and standard error was calculated:Emergence (%) = (Total emergence seeds/Total number of seeds) × 100(1)
Standard error (SE) = sqrt ((Percentage of emergence/100) × (1 − (Percentage of emergence/100))/Total number of seeds)(2)

Morphological parameters. The height (cm) of silver birch seedlings was measured after the vegetation period in one-year-old and two-year-old silver birch trees.

### 4.4. Collection of Samples for Biochemical Analyses

Leaf samples of the silver birch half-sib family seedlings were collected in three biological replicates from control and treated groups at the end of the 2020 and 2021 vegetation seasons. Samples of seedling leaves were collected by taking 3–5 leaves × 3 replicates in each group. Total number of samples per one sampling was 7 silver birch half-sib families × 1 treatment (EMF and 1 control × 3 biological replicates). Each biological replicate was measured thrice (three technical replicates).

The second sampling in 2021 was quantitatively identical. Fresh leaf samples were randomly divided into equal parts (by weight) and stored at −20 °C until biochemical analyses were performed. The amount of sample used for analysis of antioxidant enzyme activity was 200 mg, 500 mg was used for estimation of antioxidant activity and 100 mg was used for lipid peroxidation and measurements of photosynthetic pigments, secondary metabolites and total content of soluble sugars.

Quantification of activities of antioxidant enzymes, antioxidant activity, amounts of photosynthetic pigments (chlorophyll a and b, carotenoids), total polyphenols (TPC), total flavonoids content (TFC), lipid peroxidation (MDA) and total soluble sugars (TSS) was performed spectrophotometrically using a SpectroStar Nano microplate reader (BMG Labtech, Offenburg, Germany) and 96-well microplates.

### 4.5. Antioxidant Enzymes Analyses

#### 4.5.1. Preparation of Leaf Samples for Antioxidant Enzyme Tests

The extract preparation was performed according to a modified methodology [101]. The sample (200 mg) of fresh leaves was triturated with liquid nitrogen and 5 mL of extraction buffer solution was added. Extraction solution consisted of 150 mM K-phosphate buffer (pH 7.8), 1% [*v*/*v*] Triton X-100 (CarlRoth, Karlsruhe, Germany), 300 mg polyvinylpolypyrrolidone (PVPP) (Sigma-Aldrich, Oakville, ON, Canada) and 5 mM ascorbate (ASC) (Chempur, Piekary Śląskie, Poland). The samples were shaken and centrifuged (Andreas Hettich GmbH & Co. KG, Tuttlingen, Germany) for 1 h, 16,090× *g*, +4 °C. After centrifugation, the supernatant was quickly transferred to 96-microplate tubes for detection of total protein (PROT) and activities of superoxide dismutase (SOD) and catalase (CAT).

For the analyses of ascorbate peroxidase (APX), guaiacol peroxidase (POX) and glutathione reductase (GR), the supernatant was separated from the extract through Sephadex G-25 (Column PD-10, Cytiva, Gillingham, UK) columns on ice [102].

#### 4.5.2. Total Protein (PROT)

The method is based on the peptide bond reaction between the reagent and proteins, when Cu^2+^ ions are reduced to Cu^+^ [56,103]. The Biuret reagent consisted of 2% Na_2_CO_3_, 0.1 N NaOH, 1% CuSO_4_ and 2% Na-K-tartrate. The crude extract was mixed with Biuret reagent and Folin–Ciocalteu reagent (1:9 *w*/*v*). The resulting mixture was kept at room temperature for 50 min before the measurement of absorption at a wavelength of 660 nm. BSA (Bovine Serum Albumin) (>98%, Sigma-Aldrich) was used to construct the calibration curve.

The total amount of protein was expressed as micrograms of the BSA equivalent in one milliliter of crude extract (mg/mL) and calculated according to the following formula:A (mg/mL) = ((C × V)/P)/1000(3)
where A is total protein concentration (mg/mL extract); C is the concentration calculated from the calibration curve (mg/g); V is the extract volume (mL); and P is the fresh material amount (g).

#### 4.5.3. Superoxide Dismutase (SOD)

The method is based on inhibition of NBT (Nitro blue tetrazolium) reduction [104]. The solution consisted of 50 mM K-phosphate buffer (pH 7.8) [105], 0.933 mM methionine (AppliChem, Darmstadt, Germany), 75 µM NBT (VWR, Darmstadt, Germany), 2 mM riboflavin (AppliChem, Darmstadt, Germany) and 20 mM EDTA (Chempur, Piekary Śląskie, Poland). The extract was mixed with the solution and absorbance was measured at a wavelength of 550 nm. Based on the total protein concentration in the leaf tissue, SOD activity was converted to its activity in the fresh biomass (unit/mg):SOD activity (IU SOD/mg (protein) per min) = (S_c_/(50 × P_eq_ × V_e_))/t(4)
where S_c_ is the slope coefficient (the highest value of absorption-he lowest value of absorption)/time of reaction (minutes); P_eq_ is the BSA equivalent based on standard curve; V_e_ is the total extract volume; and t is the time.

#### 4.5.4. Guaiacol Peroxidase (POX)

The POX analysis was performed with filtered extracts and 30% H_2_O_2_ was used as substrate [106,107]. The buffer was consisted of 50 mM K-phosphate buffer (pH 6.5) [58,105] and 16 mM pyrogallol (Chempur, Piekary Śląskie, Poland). The extract was mixed with buffer and substrate, and the analysis was conducted at a wavelength of 430 nm. The kinetic alteration in POX (upward trajectory) was observed at intervals of 35 s. Based on the total protein concentration in the tissue, the POX activity was converted to its activity in the fresh leaf biomass (µmol/mg):POX activity (µmol oxidized pyrogallol/mg (protein) per min) = ((S_c_ × V_t_)/(0.478 × V_e_ × 2.46))/P_eq_(5)
where S_c_ is the slope coefficient (the highest value of absorption-—the lowest value of absorption)/time of reaction (minutes); V_t_ is the total sample volume; V_e_ is the total extract volume; P_eq_ is the BSA equivalent based on standard curve.

#### 4.5.5. Ascorbate Peroxidase (APX)

The APX analysis was performed with filtered extract and 30% H_2_O_2_ was used as substrate [107,108,109]. The buffer consisted of 50 mM K-phosphate buffer (pH 7.0) [58,105] and 250 mM ascorbate acid (ASC) (Chempur, Piekary Śląskie, Poland). The extract was mixed with buffer and substrate, and the analysis was conducted at a wavelength of 290 nm. The kinetic alteration in APX (downward trajectory) was observed at intervals of 35 s. Based on the total protein concentration in the tissue, the APX activity was converted to its activity in the fresh leaf biomass (µmol/mg):APX activity (µmol ASC/mg (protein) per min) = ((S_c_ × V_t_)/(0.478 × V_e_ × 2.8))/P_eq_(6)
where S_c_ is the slope coefficient (the highest value of absorption-the lowest value of absorption)/time of reaction (minutes); V_t_ is the total sample volume; V_e_ is the total extract volume; and P_eq_ is the BSA equivalent based on the standard curve.

#### 4.5.6. Catalase (CAT)

The method is based on the reaction of CAT and 30% H_2_O_2_ was used as substrate [106,107]. The buffer consisted of 50 mM K-phosphate buffer (pH 7.0) [58,105]. The extract was mixed with buffer and substrate, and the absorbance was measured at a wavelength of 240 nm. The kinetic alteration in CAT (downward trajectory) was observed at intervals of 35 s. Based on the total protein concentration in the leaf tissue, the CAT activity was converted to its activity in the fresh leaf biomass (µmol/mg):CAT activity (µmol H_2_O_2_/mg (protein) per min) = ((S_c_ × V_t_)/(0.478 × V_e_ × 39.4))/P_eq_(7)
where S_c_ is the slope coefficient (the highest value of absorption-the lowest value of absorption)/time of reaction (minutes); V_t_ is the total sample volume; V_e_ is the total extract volume; and P_eq_ is the BSA equivalent based on standard curve.

#### 4.5.7. Glutathione Reductase (GR)

The method is based on the oxidation of nicotinamide adenine dinucleotide phosphate (NADPH) by GR reducing oxidized L-glutathione (PanReac Applichem, Darmstadt, Germany) [107,108,110]. The buffer consisted of 50 mM (pH 8.00) HEPES (Sigma Aldrich, St. Louis, MO, USA), 20 mM EDTA and 5 mM NADPH (CarlRoth, Karlsruhe, Germany). The filtered extract was mixed with buffer and substrate, and the analyses were conducted at a wavelength of 340 nm. The kinetic alteration in GR (downward trajectory) was observed at intervals of 35 s. Based on the total protein concentration in the tissue, GR activity was converted to its activity in the fresh leaf biomass (µmol/mg):GR activity (µmol NADPH/mg (protein) per min) = ((S_c_ × V_t_)/(0.478 × V_e_ × 6.22))/P_eq_(8)
where S_c_ is the slope coefficient (the highest value of absorption-the lowest value of absorption)/time of reaction (minutes); V_t_ is the total sample volume; V_e_ is the total extract volume; and P_eq_ is the BSA equivalent based on standard curve.

### 4.6. Preparation of Extracts for Analysis of Antioxidant Activity

Antioxidant capacity analyses by both DPPH and ABTS assays were performed using spectrophotometrical method using Synergy HT Multi-Mode Microplate Reader (BioTek Instruments, Inc., Bad Friedrichshall, Germany). For antioxidant analysis, 500 mg of fresh leaf sample was homogenized by analytical mill (Laboratory Equipment, Staufen, Germany). Then, 75% MeOH (10 mL) was added to the homogenized samples and the resulting solution was shaken at room temperature for 24 h using a Kuhner Shaker X electronic shaker (Adolf Kühner AG, Birsfelden, Switzerland). After 24 h, leaf extracts were purified using Rotilabo^®^-113A (Ø 90 mm) filter papers (Carl Roth, Karlsruhe, Germany).

#### 4.6.1. DPPH (2,2-Diphenyl-1-picryl-hydrazyl-hydrate) Radical Scavenging

Free radical scavenging capacity in methanol extracts of silver birch leaf samples was determined spectrophotometrically by DPPH >97% (2,2–diphenyl–1–picrylhydrazyl) assay as described in [111]. The reaction solution consisted of DPPH (7.9 mg of DPPH was dissolved in 200 mL of pure methanol). The extract was mixed with DPPH solution, the mixture was left in the dark for 16 min and then the absorbance was measured at 515 nm. An equal amount of DPPH and methanol was used for a blank. Trolox was used as the standard. The radical scavenging activity was calculated as antioxidant Trolox equivalents per gram of fresh material according to Equation (7) (was modified by Xiao et al., 2020 [112]).
TE = (C × V)/M(9)
where C is the Trolox concentration (mM/mL); V is the extract volume (mL); and M is the fresh material amount (g).

#### 4.6.2. ABTS (2,2′-Azino-bis(3-ethylbenzothiazoline-6-sulfonic Acid)) Radical Scavenging

Antiradical assay of silver birch leaf samples was carried out using spectrophotometrical method by the radical cation ABTS 98% (2,2′-azino-di-[3-ethylbenzthiazoline sulphonate]) generated following [113]. The reaction solution consisted of ABTS (Fluka, Dresden, Germany), 56 mg dissolved in 20 mL H_2_O, and 70 nM K_2_S_2_O_8_. The extract (0.05 mL) was mixed with ABTS solution, the mixture was left in a dark place at room temperature for 10 min and then the absorbance was measured at 734 nm. Trolox was used as the standard. The radical scavenging activity was calculated as antioxidant Trolox equivalents per gram of fresh material according to Equation (7).

### 4.7. Preparation of Extracts for Detection of Photosynthetic Pigments, TPC, TFC, Lipid Peroxidation and TSS

The sample (100 mg) of silver birch leaves was homogenized using a Precellys homogenizer (Bertin Technologies, Montigny-le-Bretonneux, France) (5500 rpm 30 s × 2, 80% ethanol (2 mL) was added and centrifuged 4000 rpm 30 s). Then, the samples were centrifuged for 30 min, 21,910× *g*, +4 °C using a Hettich Universal 32R centrifuge (Andreas Hettich GmbH & Co. KG, Tuttlingen, Germany). After centrifugation, the supernatant was transferred to the 96-microplate tubes and used for analyses.

#### 4.7.1. Quantification of Chlorophylls a and b and Total Carotenoids

To minimize the degradation of components by light, the analysis of photosynthetic pigments was carried out in the dark using an extract of fresh leaves. The absorption of the extract was conducted at the wavelengths of 664 nm (chlorophyll a), 648 nm (chlorophyll b) and 471 nm (carotenoids). The content of chlorophyll a, chlorophyll b and total carotenoids was calculated using the formulas produced by Lichtenthaler and Buschmann [114]:C(chl a) = (13.36 × A_664_) − (5.19 × A_648_)(10)
C(chl b) = (27.43 × A_648_) − (8.12 × A_664_)(11)
C(carotenoids) = (1000 × A_471_ − 2.13 × C(chl a) − 97.64 × C(chl b))/209(12)
where A is the absorption of the extract at the respective wavelength and C(chl a), C(chl b) and C(carotenoids) is the concentrations of alpha and beta chlorophyll and total carotenoids in the extract (μg/mL).

Photosynthetic pigment concentration per one gram of fresh leaves is calculated according to the following formula:Content (µg/g) = (C × V × W)/M(13)
where the content of pigments in fresh leaves is expressed in (μg/g); C is the concentration of pigment in the extract (μg/mL); V is the volume of crude extract (mL); W is the dilution of crude extract (units); and M is the fresh material amount (g).

#### 4.7.2. Detection of Total Phenolic Content (TPC)

TPC was measured using the Folin–Ciocalteu reagent according to a modified methodology [115]. The extract was mixed with Folin–Ciocalteu reagent (1:9 *w*/*v*) (VWR International GmbH, Vienna, Austria). Samples were mixed well and incubated at room temperature for 5 min; then, 10% of sodium carbonate (Na_2_CO_3_) was added. Samples were incubated for 1 h in the dark. The absorption of the extract was measured at a wavelength of 725 nm.

Gallic acid (>98%, Carl Roth GmbH + Co. KG, Karlsruhe, Germany) was used for the calibration curve.

TPC is expressed as micrograms of gallic acid equivalent in one gram of fresh leaf mass (mg/g):Content (mg/g) = (C × V)/M(14)
where C is the content obtained from the calibration curve (mg/mL); V is the extract volume (mL); and M is the fresh material amount (g).

#### 4.7.3. Detection of Total Flavonoid Content (TFC)

TFC was determined based on the formation of a flavonoid–Al(III) complex [116]. The reaction buffer consisted of 60 μL of absolute ethyl alcohol, 10% (*w*/*v*) aluminum chloride (99% purity) solution, 1 M potassium acetate (99% purity) and 120 μL distilled water. The extract was mixed with reaction buffer and the absorption of the extract was conducted at a wavelength of 415 nm.

Quercetin (>98%, Cayman Chemical Company, Ann Arbor, MI, USA) was used for the calibration curve and TFC is expressed as micrograms of the quercetin equivalent in one gram of fresh biomass (mg/g).

The TFC can be calculated using the Equation (14).

#### 4.7.4. Detection of Lipid Peroxidation (MDA)

Lipid peroxidation was determined by measuring malondialdehyde (MDA) content using the thiobarbituric acid–malondialdehyde method [117,118,119]. The extract supernatant was mixed with 0.5% (*w*/*v*) thiobarbituric acid (TBA) dissolved in 20% (*w*/*v*) trichloroacetic acid. The reaction mixture was incubated for 1 h in a heating oven heated to 90 °C (Agro-LAB termoastating TFC 200, Venice, Italy). Absorbance of cooled samples was measured at three wavelengths: 440 nm, 532 nm and 600 nm. MDA concentration was calculated using the following formula:MDA = 6.45 · (A_532_ − A_600_) − (0.56 · A_440_)(15)
where Ax is the absorption at the corresponding wavelength.

The concentration of MDA was expressed as nmol MDA per gram of crude tissue based on dilution and sample weight.

#### 4.7.5. Total Soluble Sugars (TSS)

TSS were determined using the methodology of Leyva et al. (2008) [120]. The supernatant was mixed with 0.1% anthrone reagent (CarlRoth, Karlsruhe, Germany) and heated at 90 °C for 1 h (Agro-LAB termoastating TFC 200, Venice, Italy). The absorbance of the cooled samples was measured at a wavelength of 620 nm. Glucose is used to create the calibration curve and the amount of soluble sugars was expressed as glucose equivalents (mg) per gram of raw tissue based on dilution and sample weight.

### 4.8. Statistical Analysis

Group means, standard errors and statistical analysis were performed using Microsoft Excel. The T-TEST procedure was used for pairwise comparisons of the treatments among the treated and the un-treated samples (* *p* < 0.05; ** *p* < 0.01; *** *p* < 0.001). The same method was applied for the comparison between the ages of silver birch. In the graphs, solid colors indicate significant differences between the seedling years (one-year-old and two-year-old) in the same experimental group.

## 5. Conclusions

The obtained data indicate that seed treatment with EMFs could potentially modulate the levels of numerous biochemical compounds, as well as the activities of antioxidant enzymes (closely associated with plant stress resistance) in the leaves of silver birch seedlings. However, it is crucial to note the differences in responses that are based on the plant genotype. The obtained results provide clearer insight into a specific group of indicators involved in seedling response to EMF treatment. Stronger positive changes in the emergence, seedling growth, antioxidant system, TPC, TFC and TSS has been found in three silver birch half-sib families (60th, 73rd and 179th) out of the seven families studied. These results unveil the complexity of plants responses to stress induced by seed treatment with stressors (such as EMF or others). In addition, aside from multifactorial and systematic changes occurring at different levels of structural organization (including proteomic and metabolomic), a strong dependency on slight genetic interspecies differences becomes obvious. Considering all those previous studies together, it is imperative to conduct further studies with pathogen-infected trees in order to evaluate the effects on growth and disease resistance.

## Figures and Tables

**Figure 1 plants-12-03048-f001:**
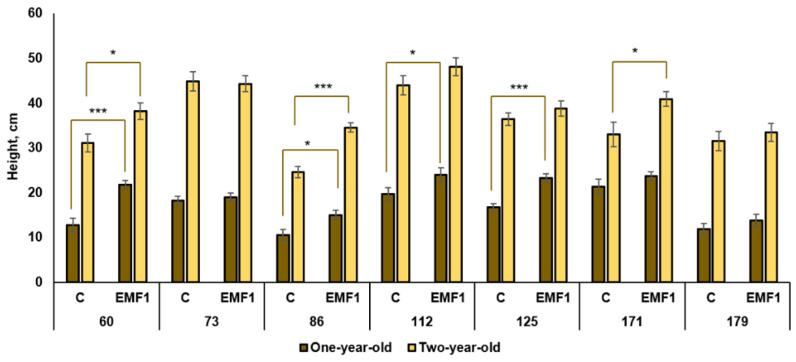
Height of silver birch seedlings growing from control (C) and EMF-treated (EMF1) seeds. The statistical significance of the difference between the treated group and the control in each half-sib family is shown by asterisks (* *p* < 0.05; *** *p* < 0.001). The brown color shows samples from one-year-old seedlings; the yellow color shows samples from two-year-old seedlings. Statistically significant differences between one-year-old and two-year-old seedling height in the same experimental group are shown by a solid column color (*p* < 0.05).

**Figure 2 plants-12-03048-f002:**
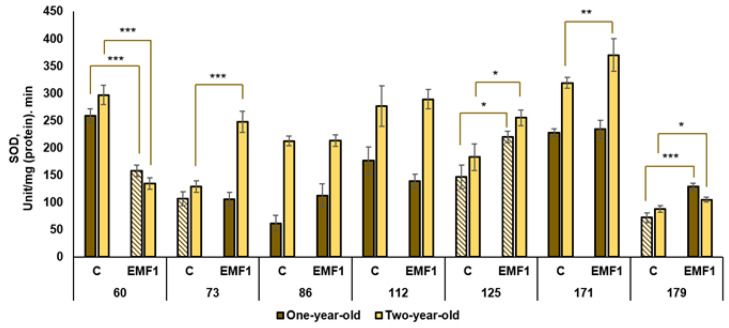
Activity of superoxide dismutase (SOD) in leaves of silver birch growing from control (C) and EMF-treated (EMF1) seeds (n = 9; 3 biological replicates × 3 technical replicates). The asterisks denote the statistical significance of the difference between the treated group (EMF1) and the control in each half-sib family (* *p* < 0.05; ** *p* < 0.01; *** *p* < 0.001). Statistically significant differences between enzyme activity in leaves of the one-year-old and two-year-old seedlings in the same experimental group are shown by a solid color column (*p* < 0.05). The brown color shows samples from one-year-old seedlings; the yellow color shows samples from two-year-old seedlings. The column filled by dash means not statistically significant between enzyme activity in leaves of the one-year-old and two-year-old seedlings in the same experimental group.

**Figure 3 plants-12-03048-f003:**
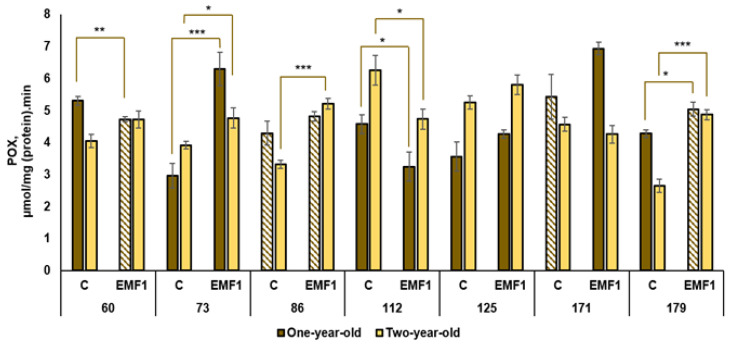
Activity of peroxidase POX in leaves of silver birch growing from control (C) and EMF-treated (EMF1) seeds (n = 9; 3 biological replicates × 3 technical replicates). The asterisks denote the statistical significance of the difference between the treated group (EMF1) and the control in each half-sib family (* *p* < 0.05; ** *p* < 0.01; *** *p* < 0.001). Statistically significant differences between enzyme activity in leaves of one-year-old and two-year-old seedlings in the same experimental group are shown by a solid color column (*p* < 0.05). The brown color shows samples from one-year-old seedlings; the yellow color shows samples from two-year-old seedlings. The column filled by dash means not statistically significant between enzyme activity in leaves of the one-year-old and two-year-old seedlings in the same experimental group.

**Figure 4 plants-12-03048-f004:**
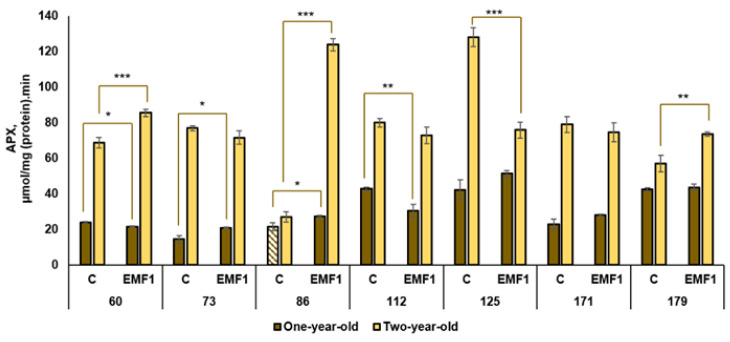
Activity of ascorbate peroxidase APX in leaves of silver birch growing from control (C) and EMF-treated (EMF1) seeds (n = 9; 3 biological replicates × 3 technical replicates). The asterisks denote the statistical significance of the difference between the treated group (EMF1) and the control in each half-sib family (* *p* < 0.05; ** *p* < 0.01; *** *p* < 0.001). Statistically significant differences between enzyme activity in leaves of one-year-old and two-year-old seedlings in the same experimental group are shown by a solid color column (*p* < 0.05). The brown color shows samples from one-year-old seedlings; the yellow color shows samples from two-year-old seedlings. The column filled by dash means not statistically significant between enzyme activity in leaves of the one-year-old and two-year-old seedlings in the same experimental group.

**Figure 5 plants-12-03048-f005:**
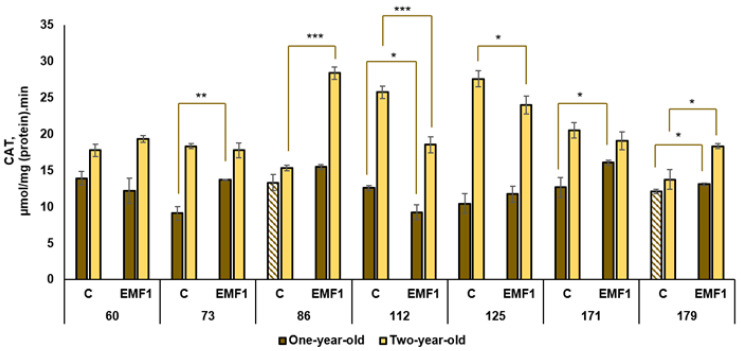
Activity of catalase (CAT) in leaves of silver birch growing from control (C) and EMF-treated (EMF1) seeds (n = 9; 3 biological replicates × 3 technical replicates). The asterisks denote the statistical significance of the difference between the treated group (EMF1) and the control in each half-sib family (* *p* < 0.05; ** *p* < 0.01; *** *p* < 0.001). Statistically significant differences between enzyme activity in leaves of one-year-old and two-year-old seedlings in the same experimental group are shown by a solid color column (*p* < 0.05). The brown color shows samples from one-year-old seedlings; the yellow color shows samples from two-year-old seedlings. The column filled by dash means not statistically significant between enzyme activity in leaves of the one-year-old and two-year-old seedlings in the same experimental group.

**Figure 6 plants-12-03048-f006:**
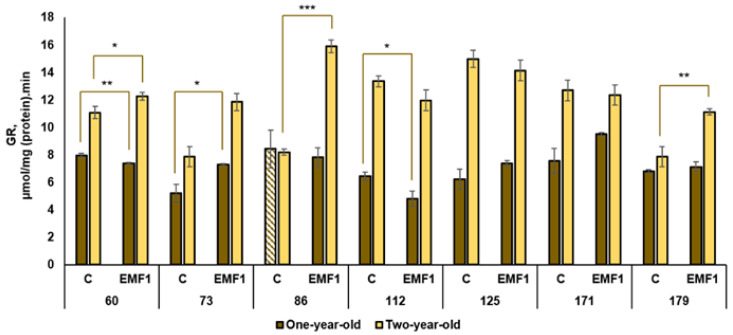
Activity of glutathione reductase (GR) in leaves of silver birch growing from control (C) and EMF-treated (EMF1) seeds (n = 9; 3 biological replicates × 3 technical replicates). The asterisks denote the statistical significance of the difference between the treated group (EMF1) and the control in each half-sib family (* *p* < 0.05; ** *p* < 0.01; *** *p* < 0.001). Statistically significant differences between enzyme activity in leaves of one-year-old and two-year-old seedlings in the same experimental group are shown by a solid color column (*p* < 0.05). The brown color shows samples from one-year-old seedlings; the yellow color shows samples from two-year-old seedlings. The column filled by dash means not statistically significant between enzyme activity in leaves of the one-year-old and two-year-old seedlings in the same experimental group.

**Figure 7 plants-12-03048-f007:**
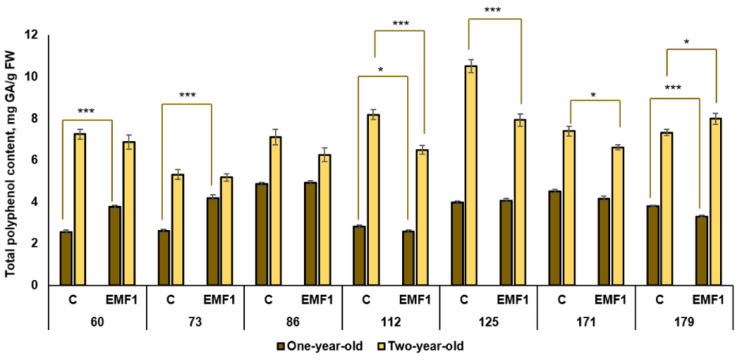
Total polyphenol content (±SE) in leaves of silver birch seedlings growing from control (C) and EMF-treated seeds (EMF1) (N = 9; 3 biological replicates × 3 technical replicates). The asterisks denote the statistically significant EMF effect in each half-sib family (* *p* < 0.05; *** *p* < 0.001). Statistically significant differences between TPC amount in leaves of one-year-old and two-year-old seedlings in the same experimental group are shown by a solid color column (*p* < 0.05). The brown color shows samples from one-year-old seedlings; the yellow color shows samples from two-year-old seedlings.

**Figure 8 plants-12-03048-f008:**
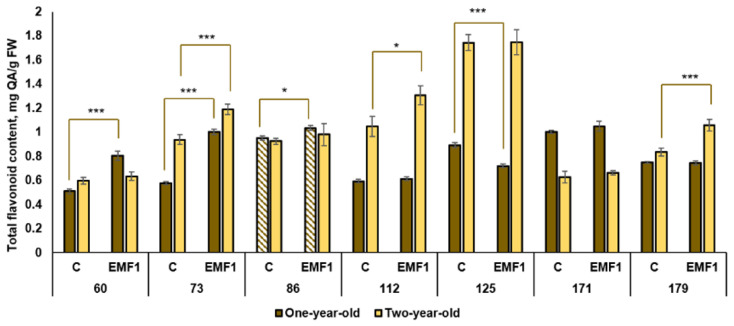
Total flavonoid content (±SE) in leaves of silver birch seedlings growing from control (C) and EMF-treated seeds (EMF1) (n = 9; 3 biological replicates × 3 technical replicates). The asterisks indicate the statistically significant effect of EMF treatment in each half-sib family (* *p* < 0.05; *** *p* < 0.001). Statistically significant differences between TFC amount in leaves of one-year-old and two-year-old seedlings in the same experimental group are shown by a solid color column (*p* < 0.05). The brown color shows samples from one-year-old seedlings; the yellow color shows samples from two-year-old seedlings. The column filled by dash means not statistically significant between TFC in leaves of the one-year-old and two-year-old seedlings in the same experimental group.

**Figure 9 plants-12-03048-f009:**
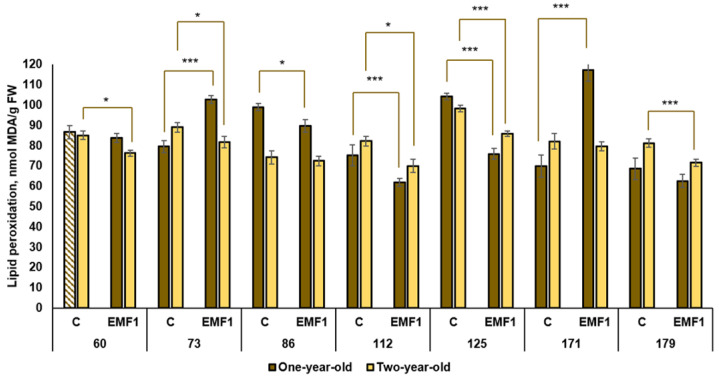
Lipid peroxidation (MDA) (±SE) in leaves of silver birch seedlings growing from control (C) and EMF-treated seeds (EMF1) (n = 9; 3 biological replicates × 3 technical replicates). The asterisks denote the statistical significance of the difference between the treated group (EMF1) and the control (C) in each half-sib family (* *p* < 0.05; *** *p* < 0.001). Statistically significant differences between MDA levels in leaves of one-year-old and two-year-old seedlings in the same experimental group are shown by a solid color column (*p* < 0.05). The brown color shows samples from one-year-old seedlings; the yellow color shows samples from two-year-old seedlings. The column filled by dash means not statistically significant between MDA in leaves of the one-year-old and two-year-old seedlings in the same experimental group.

**Figure 10 plants-12-03048-f010:**
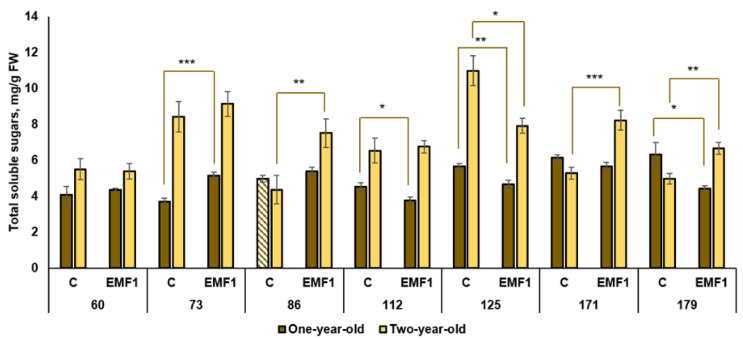
Total soluble sugar content (±SE) in leaves of silver birch seedlings growing from control (C) and EMF-treated (EMF1) seeds (n = 9; 3 biological replicates × 3 technical replicates). The asterisks denote the statistical significance of the difference between the treated group (EMF1) and the control (C) in each half-sib family (* *p* < 0.05; ** *p* < 0.01; *** *p* < 0.001). Statistically significant differences between TSS content in leaves of one-year-old and two-year-old seedlings in the same experimental group are shown by a solid color column (*p* < 0.05). The brown color shows samples from one-year-old seedlings; the yellow color shows samples from two-year-old seedlings. The column filled by dash means not statistically significant between TSS in leaves of the one-year-old and two-year-old seedlings in the same experimental group.

**Figure 11 plants-12-03048-f011:**
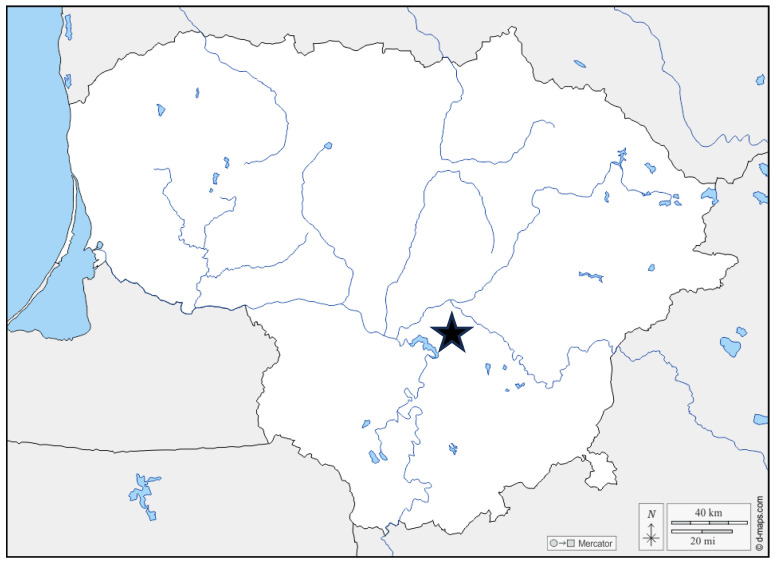
Map of Lithuania showing study site: seed orchard (marked as star on the map).

**Figure 12 plants-12-03048-f012:**
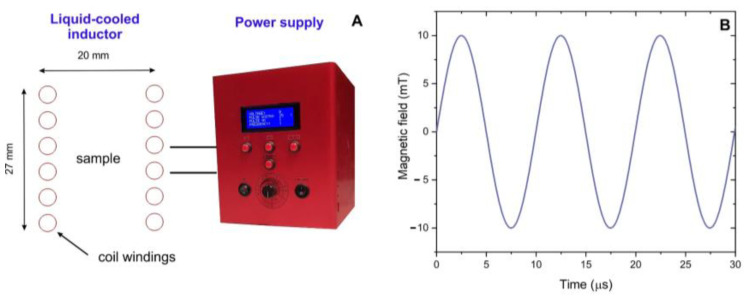
The scheme and photo of the magnetic field generator (**A**) and the representative waveform measured at maximum field intensity (**B**).

**Figure 13 plants-12-03048-f013:**
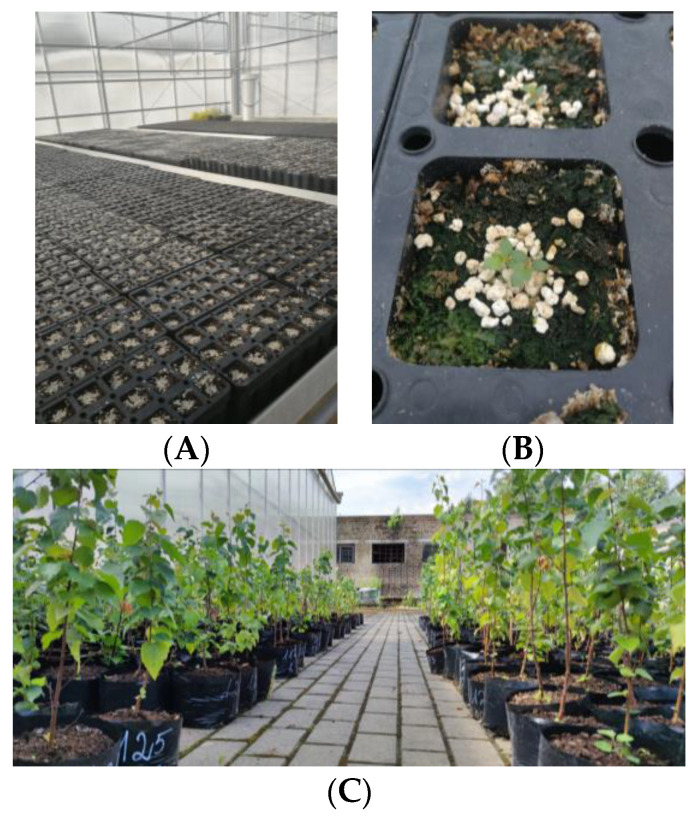
The cassettes with the sown seeds in the greenhouse (**A**). The emerged seedling of silver birch (**B**) and two-year-old silver birch trees (**C**).

**Table 1 plants-12-03048-t001:** Emergence of silver birch seedlings. C—control, EMF1—electromagnetic field 1 min. The statistical significance of the difference between the treated group and the control is shown by asterisks (** *p* < 0.01).

Half-Sib Family	Treatment	Emergence (%) ± SE
60	C	2.31 ± 0.01
	EMF1	7.87 ± 0.02 **
73	C	16.67 ± 0.03
	EMF1	17.59 ± 0.03
86	C	3.24 ± 0.01
	EMF1	3.70 ± 0.01
112	C	6.02 ± 0.02
	EMF1	11.11 ± 0.02
125	C	10.65 ± 0.02
	EMF1	6.94 ± 0.02
171	C	4.17 ± 0.01
	EMF1	7.87 ± 0.02
179	C	3.7 ± 0.01
	EMF1	4.86 ± 0.01

**Table 2 plants-12-03048-t002:** Antioxidant (DPPH and ABTS radical scavenging) activity in leaves of one-year-old and two-year-old silver birch seedlings (±SE) (N = 9; 3 biological replicates × 3 technical replicates). C—control, EMF1—electromagnetic field 1 min. The asterisks denote the statistical significance of the difference between the treated group (EMF1) and the control (C) in each half-sib family (* *p* < 0.05; ** *p* < 0.01; *** *p* < 0.001). All families had statistically significant differences between antioxidant activity in leaves of one-year-old and two-year-old seedlings in the same experimental group (*p* < 0.05).

		Radical Scavenging Activity, µmol/g, ±SE			
		One-Year-Old		Two-Year-Old	
Half-Sib Family	Treatment	DPPH	ABTS	DPPH	ABTS
60	C	331.5 ± 17.1	370.0 ± 29.7	549.2 ± 12.1	982.7 ± 31.2
	EMF1	643.2 ± 82.5 **	607.0 ± 49.4 **	540.5 ± 15.3	914.1 ± 51.2
73	C	225.0 ± 33.4	337.8 ± 24.2	551.6 ± 14.2	1163.4 ± 25.3
	EMF1	283.5 ± 14.8	546.9 ± 19.0 ***	529.5 ± 23.6	951.8 ± 19.8 ***
86	C	421.0 ± 43.5	523.9 ± 31.1	540.6 ± 11.4	1084.5 ± 17.4
	EMF1	331.8 ± 14.9	405.1 ± 18.8 **	554.9 ± 8.9	1222.3 ± 24.4 ***
112	C	346.1 ± 11.3	413.9 ± 35.5	607.0 ± 9.1	1274.9 ± 23.5
	EMF1	303.3 ± 14.8 *	382.6 ± 10.5	563.3 ± 9.9 **	1154.9 ± 19.7 **
125	C	400.3 ± 14.6	534.9 ± 28.0	596.8 ± 8.5	1341.6 ± 39.3
	EMF1	456.2 ± 20.4 *	556.5 ± 54.4	586.6 ± 11.4	1317.1 ± 25.5
171	C	1025.4 ± 14.7	686.9 ± 41.4	615.7 ± 19.7	1382.3 ± 24.2
	EMF1	322.0 ± 35.6 ***	636.5 ± 56.4	597.0 ± 15.5	1289.4 ± 17.6 **
179	C	426.4 ± 21.5	615.5 ± 23.3	612.9 ± 20.5	1125.9 ± 13.2
	EMF1	356.4 ± 20.4 *	518.4 ± 48.1	555.6 ± 27.5	1071.2 ± 19.4 *

**Table 3 plants-12-03048-t003:** Chlorophyll a (chl a), chlorophyll b (chl b) and carotenoid (caro) content (±SE) in leaves of seedlings of different half-sib families growing from control (C) and electromagnetic-field-treated (EMF1) seeds (N = 9; 3 biological replicates × 3 technical replicates). The asterisks denote the statistical significance of the difference between the treated and control seedlings in each half-sib family (* *p* < 0.05; ** *p* < 0.01; *** *p* < 0.001). The marked values mean the difference between photosynthesis pigments in leaves of the one-year-old and two-year-old seedling in the same experimental group was not significant (*p* > 0.05).

Half-Sib Family	Treatment	Photosynthetic Pigment Content (μg/g), ±SE					
		One-Year-Old			Two-Year-Old		
		Chl a	Chl b	Caro	Chl a	Chl b	Caro
60	C	208.04 ± 27.49	101.60 ± 10.78	14.49 ± 0.22	255.72 ± 21.90	216.65 ± 8.72	66.05 ± 3.44
	EMF1	325.35 ± 21.15 **	148.16 ± 8.99 ***	15.56 ± 0.15 ***	637.03 ± 26.83 ***	416.61 ± 14.54 ***	49.97 ± 1.16 **
73	C	244.58 ± 7.42	120.29 ± 3.07	15.10 ± 0.06	256.83 ± 9.64	196.18 ± 5.90	61.70 ± 2.10
	EMF1	387.62 ± 8.97 ***	180.67 ± 3.82 ***	15.27 ± 0.04 *	332.62 ± 16.16 ***	235.78 ± 12.10 ***	59.18 ± 1.77
86	C	395.81 ± 4.32	187.49 ± 1.82	14.01 ± 0.12	357.63 ± 36.08	261.32 ± 20.33	32.51 ± 1.44
	EMF1	385.26 ± 1.77 *	177.55 ± 0.35 ***	14.15 ± 0.13	583.58 ± 30.68 ***	376.02 ± 10.78 ***	35.47 ± 0.98
112	C	225.83 ± 10.07	111.34 ± 4.14	17.36 ± 0.04	285.93 ± 26.09	196.94 ± 14.61	46.86 ± 2.09
	EMF1	272.54 ± 8.50 **	126.01 ± 3.67 *	16.71 ± 0.23 *	301.53 ± 8.96	209.97 ± 5.43	49.43 ± 0.49
125	C	488.29 ± 9.46	236.13 ± 3.66	14.01 ± 0.07	371.40 ± 28.43	271.38 ± 13.60	48.08 ± 1.62
	EMF1	421.43 ± 10.86 ***	211.47 ± 4.93 ***	13.85 ± 0.11	427.17 ± 31.83	296.01 ± 16.26	40.84 ± 1.04 **
171	C	415.18 ± 14.31	199.51 ± 6.82	14.83 ± 0.23	255.92 ± 23.27	192.09 ± 11.53	48.90 ± 1.15
	EMF1	323.91 ± 50.95	160.96 ± 22.29	17.31 ± 0.58 **	350.75 ± 32.77 *	238.39 ± 14.69 *	58.27 ± 2.27 **
179	C	303.45 ± 4.10	140.95 ± 0.97	15.92 ± 0.07	435.61 ± 32.47	274.36 ± 16.62	48.40 ± 0.94
	EMF1	332.31 ± 8.86 **	158.50 ± 4.45 **	18.00 ± 0.29 ***	524.40 ± 26.20 *	320.44 ± 12.66 *	47.87 ± 0.58

**Table 4 plants-12-03048-t004:** Summary of the EMF effects observed in one-year-old (**A**) and two-year-old (**B**) silver birch seedlings: the green up arrow indicates positive effects, while the red down arrow indicates negative changes. Neutral effects are marked by zero.

**(A)** One-year-old seedlings of Silver birch							
Half-sib family	60	73	86	112	125	171	179
Emergence	↑	0	0	0	0	0	0
Growth	↑	0	↑	↑	↑	0	0
SOD	↓	0	0	0	↑	0	↑
POX	↓	↑	0	↓	0	0	↑
APX	↓	↑	↑	↓	0	0	0
CAT	0	↑	0	↓	↑	↑	↑
GR	↓	↑	0	↓	0	0	0
DPPH	↑		0	↓	↑	↓	↓
ABTS	↑	↑	↓	0	0	0	0
Chl a	↑	↑	↓	↑	↓	0	↑
Chl b	↑	↑	↓	↑	↓	0	↑
Caro	↑	↑	0	↓	0	↑	↑
TPC	↑	↑	0	↓	0	0	↓
TFC	↑	↑	↑	0	↓	0	0
MDA	0	↑	↓	↓	↓	↑	0
TSS	0	↑	↑	↓	↓	0	↓
**(B)** Two-year-old seedlings of Silver birch							
Half-sib family	60	73	86	112	125	171	179
Growth	↑	0	↑	0	0	↑	0
SOD	↓	↑	0	0	↑	↑	↑
POX	0	↑	↑	↓	0	0	↑
APX	↑	0	↑	0	↓	0	↑
CAT	0	0	↑	↓	↓	0	↑
GR	↑	0	↑	0	0	0	↑
DPPH	0	0	0	↓	0	0	0
ABTS	0	↓	↑	↓	0	↓	↓
Chl a	↑	↑	↑	0	0	↑	↑
Chl b	↑	↑	↑	0	0	↑	↑
Caro	↓	0	0	0	↓	↑	0
TPC	0	0	0	↓	↓	↓	↑
TFC	0	↑	0	↑	0	0	↑
MDA	↓	↓	0	↓	↓	0	↓
TSS	0	0	↑	0	↓	↑	↑

Effects on activities of antioxidant enzymes: SOD—superoxide dismutase; POX—peroxidase; APX—ascorbate peroxidase; CAT—catalase; GR—glutathione reductase. Changes in leaf antioxidant activity: DPPH—antioxidant activity measured by DPPH scavenging; ABTS—antioxidant activity measured by ABTS binding. Effects on amounts of biochemical compounds: Chl a—chlorophyll a; Chl b—chlorophyll b; Caro—carotenoids; TPC—total phenolic content; TFC—total flavonoid content; MDA—malondialdehyde; TSS—total soluble sugars.

## Data Availability

Data will be made available upon request.

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
