# Peer review of "Seed Treatment with Electromagnetic Field Induces Different Effects on Emergence, Growth and Profiles of Biochemical Compounds in Seven Half-Sib Families of Silver Birch"

_plants, 2023, doi:10.3390/plants12173048_

Round 1
Reviewer 1 Report
I would like to extend my sincere gratitude for allowing me to review the paper titled “Stimulation of germination, growth and defensive capacity in different half-sib families of Silver birch by using seed treatment with electromagnetic field” by ČėsnienÄ— et al. I believe that this draft needs Minor correction to be accepted. My points of view are listed as follows:
Comments/ Questions
- The abstract should be rewritten. Methods of the experiment should be added in the abstract.
- The abstract doesn’t have the quantitative data which confirmed the effectiveness of EMF.
- Authors mentioned “seven different half-sibs of Silver birch” whereas this tree is a cross-pollinating plant. How can the authors be certain about the parentage of the half-sib plants?
- Authors mentioned that the seeds were treated for 1 min and then after 4 days of treatment were sown. Would you please clarify exactly the period of treatment, the interval time, and the details about other conditions? It's good if authors describe all the steps of experiments in detail which will help readers to repeat the experiment.
- I would like to suggest authors add some pictures regarding the treatment process with EMF.
- I would like to suggest authors to replace pictures 8B and C, showing the plants and adding a bar in the picture.
- Please add the full name of the abbreviated words under the tables.
- Please write the full name of all the abbreviated words in the first place in the text.
- I believe that the references in the text should be re-styled based on the journal’s format.
can be improved
Reviewer 2 Report
Authors presented the result of a comprehensive studies of the effect of EMF treatment of silver birch seeds on the production of a number of secondary metabolites including those associated with the plant defense against pathogens. It was interesting to me to read this paper, since I also work with the effect of EMF treatment on the plant development and disease resistance.
The main questions to the obtained results are the following.
1. Authors compared birch seedlings 1 and 2 years after the treatment. I work with agricultural crops, and such seed treatment with EMF fields usually has a time-limited effect measured by several months, after which the difference between the treated and control plants decreases and becomes insignificant. Probably, there is a different situation with trees, but it would be interesting to know, if the authors made some additional measurement in a shorter time scale (say, each several months after treatment)? Are authors sure the revealed difference is caused by the seed treatment rather than any other factors, which could appear during this long time passed from the treatment? As far as I read various publications dedicated to the effects of EMF treatments, the accompanying effects were evaluated mainly in relation to germination rate and various parameters measured at the early development stages. Of course, some of the positively changes parameters, say, the size of seedlings or the content of pigments, sugars, etc., could reflect the initial growth stimulation in EMF-treated seeds; since they developed more rapidly, they should be larger. However, it would be strange if a plant would response to a short-term seed treatment for many months and years; such response should inevitably damp with time.
2. The second issue is the choice of the treatment duration. One minute is not too much taking into account the field intensity, which is not too high. How authors substantiated this choice? I did not see any explanations in the text.
3. Authors included into this study the effect of EMF treatment on the content of various secondary metabolites and enzymes involved into antioxidative systems and response to pathogen-caused stress. In my opinion, the obtained data do not reflect the stated tasks, since the real improvement of biotic stress resistance of plants should be determined under conditions of artificial inoculation with a pathogen. Under such conditions one could estimate the real response to biotic stress and its changes caused by the effect of any treatments. Why authors did not arrange such experiments? The data on the antioxidant enzymes and metabolites presented in the manuscript are hardly interpreted in any systematic way, since variation in responses is significant between the variants, families, and years. This is quite understandable taking into account a large time interval passed from the treatment, and different genotypes of birch used in the study (since different genotypes are really differently react to EMF). However, in the case of artificial inoculation with pathogen, authors probably would obtain more clear and unidirectional data, which would provide any clear conclusions. In my opinion, the content of the study does not correspond to the "defensive capacity" part of the title, so I would recommend authors to change the title of the manuscript to provide more clear reflection of what has been really done.
In relation to the results, there are a lot of diagrams, which adequately reflect the obtained data, but are difficult for a complex understanding in relation to the trends in the revealed effects. I would suggest the authors could try to make a table or tables (for each year of study) combined all these results at least in a qualitative manner. Say, the table for the first year, columns for families and rows for the revealed effects manifested via % of the control (positive (+) or negative (-) changes) with designation of significantly differing values (say, in bold). Such general table would provide the better view on the revealed changes. The similar table could be done for the second year.
In general, the paper presents a large volume of work, and obtained results can be interesting for readers. However, in my opinion, prior publication of this paper authors should include answers on the three above-mentioned questions into the text for better substantiation of the performed study.
The minor comments are below.
Abstract
Line 25: please, give the common names of TPC and TFC, not only abbreviations.
Line 34: it seems the last word in this row ("field") is not necessary.
Intro
Line 99-100: please, check the sentence carefully and correct: "early successional tree species widely distributed and economically important deciduous tree..."
Results
Line 189-190: this is a quite logical trend to a time-dependent decrease the effect of the seed treatment with EMF.
Materials and Methods
Line 859: why and how did authors choose the treatment duration?
Lines 957, 971, etc.: how did you determine the slope coefficient?
Line 1034-1035: please, check and correct the sentence since the word "using" was used two times. I suggest to replace the first one with "by".
Lines 1045-1047: please, clarify the sentence, since it is unclear, if the sample was homogenized (1) birch leaves only at 5500 rpm for 2x30 s, with the further addition of 2% 80% ethanol and homogenization at 4000 rpm for 30 min, or (2) there were two homogenization stages (one in ethanol) followed by the ethanol addition (2 mL) and the further centrifugation.
Line 1050: I consider the word "removed" should be replaced with "collected" or "transferred" into 96-well plates.
Line 1054: it would be good to explain, which wavelength was used for which pigment. Though this information presents in the formulas, it should be also added in the text.
Line 1057: please, check and correct the sentence.
English language is quite good and understandable, though I would recommend some language editing prior publication.
Reviewer 3 Report
1. The title of the publication is not adequate; the work does not present the effect on seed germination, buton seedling emergence. The use of the term "defensive capacity" is too strong. Possibly, we can only talk about the assessment of metabolic indicators that may potentially affect defense capacity. 2. Abstract. Again, "seed germination" was used instead of seedling emergence. Too little can be learned from the summary. The first, general part of this chapter is too extensive.” bolstering their resistance against pathogens "-this was not studied in the paper-the resistance to pathogens was not tested. So one can only consider a potential increase taking into account examples, from other works, this should be included in the discussion. "biologically active compounds in trees" - in trees? - too general. TPC, TFC - no full names. “Assessment of tree defense capacity was evaluated in treated and control seeds in one-year-old and two-year-old tree seedlings” – awkward wording. You can only find out / one sentence/ that the treatment affected various parameters from another that it increased the content, and what about enzyme activity? Last sentence is too strong. In summary, the abstract must be written correctly according to the rules. What is new in this research? ​3.Introduction.There is a lack of information about the role of determined markers of ISR defense. There is also a lack of information about Betula pathogens. In my opinion systemic resistance can not be considered without introducing a pathogen in the story. 4. Results. Need improvement.SD is used, not SE. The Bradford method is presently commonly used for protein determination. Titles can be improved eg. Fig.1. The chapter should be shortened more imported results. data should not repeated in the text but can compared, and describe only; authors use unnecessary:e.g."The results showed,Our results showed, Results showed". Only statistically significant results can be considered, so do not use often "significant".5.Discussion. The main question for the authors, what is new in their results compared to those obtaine by other researches cited in the discussion. Discussion is too long. L. 701-711 is introduction. L.712-714 and L.725-726 it is the same information. "Pre-treatment seed induction" is wrong formulation. "Seed emergence"- is Ok??. Results are often repeated but should be compared with others data. L 810-813 can be used in abstract and/or at the end of chapter. Discussion must re-write, is not clear, confused there are elements of introduction and results. 6.Conclusion. Again authors using seed induction. Authors did not examined mechanism of resistance, it is unknown whether determined changes are responsible to pathogenes. The mechanism of resistance was not studied only the effect of seed treatment on potential ISR markers of resistance.
Round 2
Reviewer 2 Report
Authors provided detailed comments and explanations to all my questions, and modified the manuscript in accordance with them. I have no serious questions to this new version of the manuscript. Hope to see your next publications with the effect on direct inoculation of treated plants with pathogens, it should be interesting...
I consider authors made some language editing, so the further editing during the manuscript preparationto publication will be enough.
Reviewer 3 Report
Now is OK.